# Decomposition Methods for Machine Learning with Small, Incomplete or Noisy Datasets

**Cesar Federico Caiafa** [1,2,*], **Jordi Solé-Casals** [3,*], **Pere Marti-Puig** [3], **Sun Zhe** [4] **and Toshihisa Tanaka** [5]

[1] Instituto Argentino de Radioastronomía—CCT La Plata, CONICET/CIC-PBA/UNLP, 1894 V. Elisa, Argentina

[2] Tensor Learning Team—Center for Advanced Intelligence Project, RIKEN, Tokyo 103-0027, Japan

[3] Data and Signal Processing Research Group, University of Vic-Central University of Catalonia, 08500 Vic, Catalonia, Spain; pere.marti@uvic.cat

[4] Computational Engineering Applications Unit, Head Office for Information Systems and Cybersecurity, RIKEN, Wako-Shi 351-0198, Japan; zhe.sun.vk@riken.jp

[5] Department of Electrical and Electronic Engineering, Tokyo University of Agriculture and Technology, Tokyo 184-8588, Japan; tanakat@cc.tuat.ac.jp

[*] Correspondence: ccaiafa@fi.uba.ar (C.F.C.); jordi.sole@uvic.cat (J.S.-C.)

**Abstract:** In many machine learning applications, measurements are sometimes incomplete or noisy resulting in missing features. In other cases, and for different reasons, the datasets are originally small, and therefore, more data samples are required to derive useful supervised or unsupervised classification methods. Correct handling of incomplete, noisy or small datasets in machine learning is a fundamental and classic challenge. In this article, we provide a unified review of recently proposed methods based on signal decomposition for missing features imputation (data completion), classification of noisy samples and artificial generation of new data samples (data augmentation). We illustrate the application of these signal decomposition methods in diverse selected practical machine learning examples including: brain computer interface, epileptic intracranial electroencephalogram signals classification, face recognition/verification and water networks data analysis. We show that a signal decomposition approach can provide valuable tools to improve machine learning performance with low quality datasets.

**Keywords:** empirical mode decomposition; machine learning; sparse representations; tensor decomposition; tensor completion

## 1. Introduction

Machine learning (ML) has been developing without a break since its beginning in the middle of the 20th century with the introduction of the first computers. ML comprises the design and study of algorithms that can automatically learn form observations and take optimal decisions or provide valuable outputs. With recent accelerated improvements in computing power and the availability of massive datasets, ML methods based on deep neural networks, usually referred as deep learning [1], gave rise to an Artificial Intelligence (AI) revolution. AI continues changing our daily lives contributing with extraordinary advances in scientific data analysis and new technological applications [2].

ML algorithms are based on the mathematical modelling of variables and their interaction mechanisms that can explain the observations (dataset). Complex datasets, such as natural images, speech or brain signals, usually requires sophisticated ML models to capture the probability distribution of data. Most sophisticated ML models are built upon feed-forward deep neural networks having from dozens to hundreds layers leading to a very large set of model parameters. To train

such big deep learning models requires accessing to extremely large datasets, which are not always available or they are too expensive to obtain.

Standard training algorithms for modern deep learning models assume that datasets are "infinite", i.e., they are large enough to allow successful training of very large models. In practice, and particularly when an image dataset is not large enough, it is common practice to artificially generate additional samples by applying a composition of random class-preserving transformations on available data samples such as crops, translations and rotations, which is widely known as "data augmentation".

Moreover, available ML algorithms not only assume "infinite" datasets, they were also designed for perfect input data samples. Nevertheless, in practical applications data samples often suffer from imperfections, such as missing or noisy features. For example, when recording electroencephalographic (EEG) signals, corrupted data can be originated from impedance mismatching, electrode disconnection, body movements, etc. [3]. Other practical problems where data samples can be incomplete include: computer vision systems where objects in the view field can be partially occluded [4]; recommendation systems built from the information gathered by different users where not all the users have fully completed their forms [5]; or medical datasets where typically not all tests can be performed on all patients [6].

In this article, we review recent techniques to alleviate the serious consequences of having different types of low-quality datasets in ML applications. We demonstrate that, by using decomposition methods, we can model one-dimensional and multi-dimensional signals, which allow us to artificially generate class-preserving new signals or make inference on missing/corrupted features.

This article is organized as follows: in Section 1.1, a review of the state-of-the-art approaches for low-quality datasets in ML is given; in Section 1.2, the mathematical notation used throughout the paper is introduced; in Section 2, a unified view of signal decomposition methods is presented, which includes: subspace approximation, Empirical Mode Decomposition (EMD), sparse representations and tensor decompositions; Section 3 covers practical applications including ML problems in neurosciences, face detection/classification and analysis of water networks data; finally, the main conclusions and discussion are presented in Section 4.

## 1.1. ML with Low Quality Datasets: State-of-the-Art and Recent Progress

In this paper, we focus on the following types of low-quality datasets: (1) small, and (2) having incomplete or corrupted samples. The following subsections provide an overview of current approaches and recent progress in artificially generating new training samples (Section 1.1.1), and how to deal with incomplete datasets (Section 1.1.2).

### 1.1.1. Classical Data Augmentation

Artificial generation of training samples for machine learning has been used for many years, for example, in the form of virtual examples for training support vector machines in supervised learning [7,8]. In these papers, training data is augmented so that the learned model will be invariant to known transformations or perturbations. By using this technique, in [8], it was reported the lowest test error (0.6%) until that moment in 2002 on the well-known MNIST digit recognition benchmark task. Since then, data augmentation has been considered essential for the efficient training of neural networks, specially on images where it is typically performed in an ad-hoc manner by using class preserving transformations such as random cropping and rotations. Data augmentation is crucial to achieve nearly all state-of-the-art results, for example, in 2010 a new record on the best test error on MNIST dataset was reported (0.35%) by using deep neural networks [9]. Data augmentation is also fundamental to attain very good performance results on discriminative unsupervised feature learning based on convolutional neural networks [10]. While it is usually easy for domain experts to specify the involved transformations, for example the cropping and rotations in images, applications in other domains may require a non trivial choice of transformations. Motivated by this, in [11], the authors proposed a method for automating data augmentation by learning a generative sequence model

over user-specified transformation functions using a generative adversarial approach, which was successfully applied to image and text datasets.

Data augmentation was also applied in machine applications other than image classification, for example, in music source separation [12]. In that paper, the authors augmented the training datasets by randomly swapping left/right channels for each instrument, chunking into sequences for each instrument and mixing them from different songs sources, which demonstrated to boost the performance of deep neural networks for this task. Other application domains where data augmentation improved the discriminative power of classifiers include: biometrics [13], fault diagnosis in industry [14], radio frequency fingerprint identification [15], synthetic aperture radar (SAR) target recognition [16], and others.

From the theoretical point of view only very recently some understanding of the underlying theoretical principles involved in data augmentation procedure was provided. In [17], the authors provide a general model of augmentation as a Markov process and show that, combined with a $k$-nearest neighbor ($k$-NN) rule, is asymptotically equivalent to a kernel classifier. This result provides novel connections between prior work on invariant kernels, tangent propagation and robust optimization, giving an illustrative view on how augmentation affect the learning model.

### 1.1.2. Classical Approaches to ML with Incomplete Data

The classical approach to supervised learning with missing or noisy data is to preprocess the available data in order to infer missing/corrupted values such that standard ML algorithms can be used on the corrected dataset [18,19]. This imputation aproach can be based on statistical principles, such as computing the mean of available samples for missing features or more sophisticated estimators like the regression imputation, which has the advantage that it can take into consideration the correlations among various features. Other imputation methods are based on machine learning ideas by estimating missing entries through the $k$-nearest neighbor [20], Self Organization Maps (SOM) [21], multilayer or recurrent neural networks [22,23], and others.

A different approach is to avoid direct imputation of lost inputs and rely on a probabilistic model of input data, based for example on the Gaussian Mixture Model (GMM) and learning model parameters through the Expectation Maximization (EM) algorithm and building a Bayesian classification. The advantage of this approach is that class labels of input data is fully exploited which helps for a correct imputation of missing entries [19,24]. However, the latter "model-based" approach has the disadvantage that it requires a good probabilistic data model, which is usually not available, especially for real-world applications such as those involving natural images.

Recently, some approaches based on the low-rank property of the features data matrix were investigated and algorithms for data completion were proposed incorporating the label information [25–27]. Since the rank estimation of a matrix is a computationally expensive task, usually based on the Singular Value Decomposition (SVD), the obtained algorithms are prohibitive to solve modern machine learning problems with large datasets. Additionally, as in the case of the probabilistic generative models, none of these methods considered complex classifications functions. To overcome this drawback, more recently, a framework based on neural network architectures such as autoencoders, multilayer perceptrons and Radial Basis Function Networks (RBFNs), was proposed for handling missing input data by setting a probabilistic model, e.g., a GMM, for every missing value, which is trained together with the NN weights [28]. This method combined the great capability of NNs to approximate complex decision functions with the nice formulation of the GMM to model missing data. However, it inherited the drawbacks of GMMs, e.g., they are not well suited to higher-dimensional datasets.

### 1.2. Mathematical Notation and Definitions

Vectors and matrices are denoted using boldface lower- and upper-case letters, respectively. For example $\mathbf{x} \in \mathbb{R}^I$ and $\mathbf{A} \in \mathbb{R}^{I \times J}$ represent a vector and a matrix, respectively. Columns of a matrix $\mathbf{A} \in \mathbb{R}^{I \times J}$ are referred as vectors $\mathbf{a}_j \in \mathbb{R}^I$.

A tensor is a multidimensional array generalizing vectors and matrices to higher number of dimensions. For example, a tensor $\underline{\mathbf{X}} \in \mathbb{R}^{I \times J \times K}$ is a 3D array of real numbers whose elements $(i, j, k)$ are referred to as $x_{ijk}$. The individual dimensions of a tensor are referred to as modes (1st mode, 2nd mode, and so on). By generalization of matrix multiplication, a tensor can be multiplied by a matrix in a specific mode, only if their size matches. Given a tensor $\underline{\mathbf{X}} \in \mathbb{R}^{I_1 \times I_2 \cdots \times I_N}$ and a matrix $\mathbf{A} \in \mathbb{R}^{J \times I_n}$, the mode-$n$ product $\underline{\mathbf{Y}} = \underline{\mathbf{X}} \times_n \mathbf{A} \in \mathbb{R}^{I_1 \times \cdots \times I_{n-1} \times J \times I_{n+1} \cdots \times I_N}$ is defined by: $y_{i_1 \cdots i_{n-1} j i_{n+1} \cdots i_N} = \sum_{i_n=1}^{I_n} x_{i_1 \cdots i_n \cdots i_N} a_{j i_n}$, with $i_k = 1, 2, ..., I_k$ $(k \neq n)$ and $j = 1, 2, ..., J$.

The $\ell_0$-norm $\|\mathbf{x}\|_0$ of a vector $\mathbf{x} \in \mathbb{R}^N$ is defined as the number of non-zero entries of the vector. When the number of non-zero entries is much less than the dimension of the vector, i.e., $\|\mathbf{x}\|_0 \ll N$, the vector is sparse.

Given two matrices: $\mathbf{A} \in \mathbb{R}^{I_1 \times J_1}$ and $\mathbf{B} \in \mathbb{R}^{I_2 \times J_2}$, the Kronecker product is defined as follows:

$$\mathbf{A} \otimes \mathbf{B} = \begin{bmatrix} a_{1,1}\mathbf{B} & a_{1,2}\mathbf{B} & \dots & a_{1,J_1}\mathbf{B} \\ a_{2,1}\mathbf{B} & a_{2,2}\mathbf{B} & \dots & a_{2,J_1}\mathbf{B} \\ \vdots & \vdots & \ddots & \vdots \\ a_{I_1,1}\mathbf{B} & a_{I_1,2}\mathbf{B} & \dots & a_{I_1,J_1}\mathbf{B} \end{bmatrix}. \tag{1}$$

## 2. Methods

The idea of extracting valuable information from a dataset by decomposing each of the signals (data samples) as a sum of simpler components, is a very well established technique originated in branches of mathematics such as functional analysis and statistics. The general idea behind any decomposition method is to obtain a compact model that can capture the essential information of a signal or dataset. This compression capability will allow us, for example, to generate artificial data by adapting individual components, or recombining them in a different way, and using the decomposition as a generative model. On the other side, when data is incomplete, we can fit a decomposition model such that the available information is replicated as much as possible and we can use the model to estimate the values of missing data points.

The goal of this article is to present a unifying view of several useful decomposition methods and illustrate about its applications to practical ML problems. In the following, we present a mathematical formulation of the decomposition methods that will be used through the paper.

### 2.1. A unified View of Data Decomposition Models for ML

Given a linear subspace $\mathcal{U} \subset \mathbb{R}^N$, every vector data sample (one-dimensional signal) $\mathbf{x} \in \mathcal{U}$ is decomposed into a sum of $I$ components if it can be written as:

$$\mathbf{x} = \sum_{i=1}^{I} \alpha_i \boldsymbol{\phi}_i + \mathbf{r} = \boldsymbol{\Phi}\boldsymbol{\alpha} + \mathbf{r}, \tag{2}$$

where $\alpha_i \in \mathbb{R}$, the set of vectors $\{\boldsymbol{\phi}_i \in \mathbb{R}^N\}$ $(i = 1, 2, \ldots, I)$ and $\mathbf{r} \in R^N$ are the coefficients, the generators of the linear subspace $\mathcal{U}$, and the residual or approximation error, respectively. A matrix notation is also shown in the rightmost part of Equation (2), where $\boldsymbol{\Phi} \in \mathbb{R}^{N \times I}$ contains vectors $\boldsymbol{\phi}_i$ as columns and $\boldsymbol{\alpha} \in \mathbb{R}^I$ is the vector of coefficients. In some classical decomposition methods, vectors $\boldsymbol{\phi}_i$ are constructed by theoretical principles as it is the case of the discrete Fourier, Cosine or Wavelet orthogonal bases, just to mention few, where the number of components equal to the space dimension $(I = N)$, meaning that those bases span the whole space $\mathbb{R}^N$.

Given a dataset composed of $J$ vector samples $\mathbf{x}_j \in \mathbb{R}^N$, $j = 1, 2, \ldots, J$, by arranging them as columns of matrix $\mathbf{X} \in \mathbb{R}^{N \times J}$, we can write the following matrix factorization equation (see Figure 1a):

$$\mathbf{X} \approx \mathbf{\Phi} \mathbf{A}, \tag{3}$$

where $\mathbf{\Phi} \in \mathbb{R}^{N \times R}$ and $\mathbf{A} \in \mathbb{R}^{R \times J}$ has entries $\alpha_{i,j}$. Such a compact representation of datasets can be accomplished in several ways. For example, by using a subspace of a lower dimension ($R < N$) or using a sparse matrix $\mathbf{A}$ meaning that each of the signals are approximated using few non-zero coefficients, compared to the size of the space $N$. In the following subsections we describes these possible models.

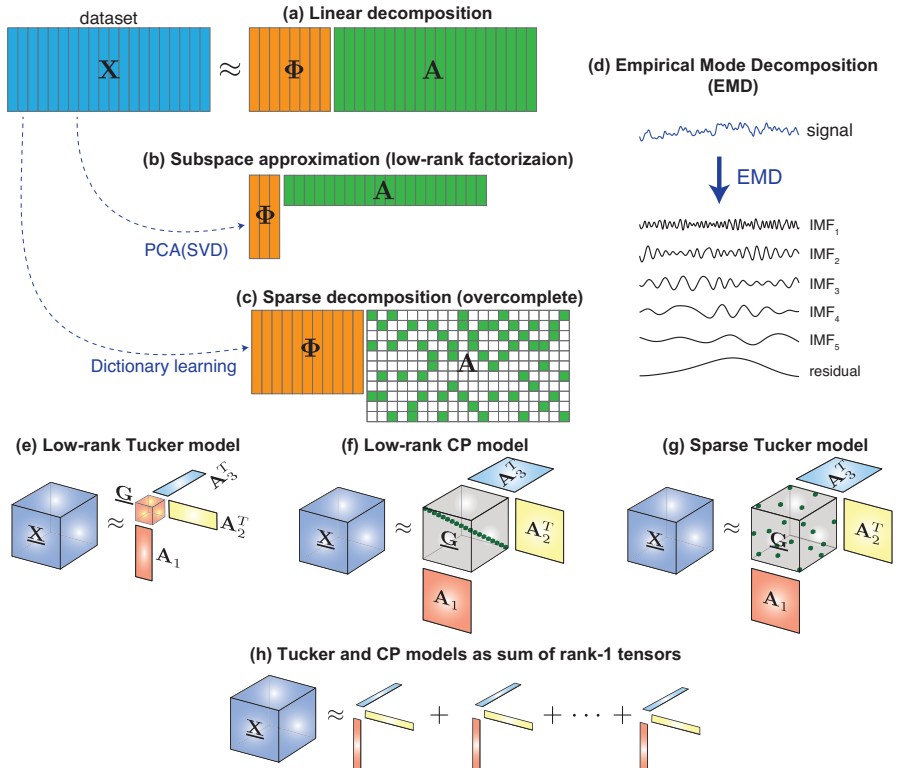

**Figure 1.** Decomposition models. (**a**) **General linear model**: a collection of vector data samples organized as columns of a matrix dataset $\mathbf{X}$ is approximated by the product of matrices $\mathbf{\Phi}$ and $\mathbf{A}$. (**b**) **Subspace** approximation: all vectors in the dataset are approximated by linear combination of few vectors (principal components). (**c**) **Sparse coding**: each vector in the dataset is approximated by the linear combination of atoms (columns of a dictionary $\mathbf{\Phi}$). In both, (**b**,**c**), the optimal choice of matrix $\mathbf{\Phi}$ can be computed from the dataset itself by means of the SVD and an dictionary learning algorithm, respectively. (**d**) **EMD**: every single signal is decomposed as a sum of characteristic modes. Tensor decomposition models such as **Low-rank Tucker** (**e**), **Low-rank CP** (**f**) and **Sparse Tucker** (**g**) can be written as sum of rank-1 tensors (**h**).

*2.2. Subspace Approximation (PCA)*

If signals in a dataset can be well approximated within a subspace of lower dimension ($R < N$), we can find the optimal basis by applying the celebrated Principal Component Analysis (PCA) algorithm. This method was originally introduced in statistics by Pearson in 1901 [29], but developed later independently by Karhunen [30] and Loéve [31]. PCA basis vectors $\boldsymbol{\phi}_i$ and their associated coefficients are easily computed by means of a Singular Value Decomposition (SVD) of the data covariance matrix. Formally, given a set of normalized signals $\{\mathbf{x}_j \in R^N\}$ (zero-mean and unit-variance samples $n = 1, 2, \ldots, J$), the rank-$R$ PCA decomposition of any signal $\mathbf{x}_j$ in the set is obtained by Equation (2), with orthonormal vectors $\boldsymbol{\phi}_i$ corresponding to the first $R < N$ dominant singular vectors

of the data covariance matrix and coefficients are computed by $\alpha_{i,j} = \mathbf{x}_j^T \boldsymbol{\phi}_i$. In this case, the obtained decomposition model is also referred as low-rank matrix factorization as illustrated in Figure 1b.

## 2.3. Sparse Decomposition (SD)

More recently, in the signal processing field, it was discovered that a better way to capture the structure of natural images, speech audio an other types of signals is to have available a large collection of prototype atoms $\{\boldsymbol{\phi}_i \in \mathbb{R}^N\}$ ($i = 1, 2, \ldots, I$) with $I \geq N$ and use only few and distinctive coefficients to represent every signal $\mathbf{x}$ in the space. This model is mathematically described by adding an sparsity constraint $\|\boldsymbol{\alpha}\|_0 \leq K$ to the model of Equation (2) leading to the following equation:

$$\mathbf{x} \approx \boldsymbol{\Phi}\boldsymbol{\alpha}, \text{ with } \|\boldsymbol{\alpha}_i\|_0 \leq K \ll N, \tag{4}$$

This approach is usually referred as "sparse coding" or "sparse representation" of signals and matrix $\boldsymbol{\Phi} \in \mathbb{R}^{N \times I}$ is called a "dictionary" [32,33] (see Figure 1c).

For general overcomplete ($I \geq N$) dictionaries, the sparse vector of coefficients can be obtained by applying some of the available algorithms that were designed to solve the sparse coding problem, which includes greedy methods such as matching pursuit (MP) [34], orthogonal matching pursuit (OMP) [35], compressive sampling matching pursuit (CoSaMP) [36], $\ell_1$ norm minimization methods such as basis pursuit [37] and many others (see [38] for a review of algorithms).

Some theoretically derived dictionaries, e.g., those based on the Discrete Cosine Transform (DCT) or Wavelet Transform (WT), are excellent candidates for sparse coding. However, in some ML applications when large datasets are available, sometimes it is good idea to learn an optimal dictionary for an specific dataset. To that end, some dictionary learning algorithms are proposed in the literature [39,40]. Sparse coding has to date a large record of successful applications in signal processing tasks like compressed sensing [41,42], blind source separation [43], denoising [39], inpainting [44] and others.

## 2.4. Empirical Mode Decomposition (EMD)

In the previously introduced methods (PCA and SD), one set of vectors $\{\boldsymbol{\phi}_i\}$ is used to generate every signal $\mathbf{x}$ in the dataset, meaning that the chosen generators are optimal for a particular dataset. Other methods have been proposed to find signal specific set of components. This was the case of the Empirical Mode Decomposition (EMD), which was first described by Huang N. et al. in [45] as a new method to analyze nonlinear and non-stationary signals. EMD is a data-based approach that decomposes any signal into a sum of so-called Intrinsic Mode Functions (IMF) plus a residual as illustrated in Figure 1d. Therefore, the original signal is modelled as a linear combination of amplitude and frequency modulation (AM-FM) functions. Every single function (IMF) is capturing specific information from a different frequency band present in the signal and is obtained in an iterative sifting procedure. Other variants have been defined since the introduction of EMD, but they all share the basic steps presented below. Let us suppose that we want to decompose a signal $x(t)$ by means of EMD. Then its decomposition will be calculated as follows:

1.  Determine the local maxima and minima of the signal $x(t)$.
2.  Calculate the upper (lower) envelope by interpolating the local maxima (minima) points. The interpolation can be carried out in different ways (linear interpolation, spline interpolation, etc.), which could lead to slightly different results.
3.  Calculate the local mean $m(t)$ by averaging the upper and lower envelopes.
4.  Calculate the first IMF candidate $h_1(t) = x(t) - m(t)$.
5.  Checks whether candidate $h_1(t)$ meets the criteria to be an IMF:

    *   If $h_1(t)$ meets the criteria, define the first IMF as $c_1(t) = h_1(t)$.
    *   If $h_1(t)$ does not meet the criteria, set $x(t) = h_1(t)$ and repeat from step 1



The next IMF will be extracted using the same procedure on the signal $r_1(t)$ that remains after subtracting the first IMF from the signal: $r_1(t) = x(t) - c_1(t)$. The process stops when two consecutive IMFs are (almost) identical and the empirical mode decomposition of the signal $x(t)$ is written as:

$$\mathbf{x}(t) = \sum_{i=1}^{n} c_i(t) + r_n(t), \tag{5}$$

indicating that the original signal $x(t)$ has been decomposed in $n$ IMFs plus a residual signal. This residual signal captures the trend (or the mean) of the original signal.

### 2.5. Tensor Decomposition (TD)

Sometimes input data samples have a multidimensional structure or it is useful to arrange one-dimensional signals into multidimensional arrays or tensors. For example, EEG signals are simultaneously recorded with multiple sensors (electrodes) thus, for each subject, a (time $\times$ channel) matrix is recorded. A natural way to construct a tensor for an EEG experiment is to use a third dimension to index subject, which results in a three dimensional data tensor $\underline{\mathbf{X}} \in \mathbb{R}^{I_1 \times I_2 \times I_3}$, where $I_1, I_2, I_3$ correspond to numbers of time samples, channels (sensors) and subjects, respectively.

Matrix factorization models, such as PCA in Equation (3), can be generalized to tensors by means of the Tucker($R_1, R_2, R_3$) decomposition [46]. Given a data tensor $\underline{\mathbf{X}} \in \mathbb{R}^{I_1 \times I_2 \times I_3}$, it can be decomposed as:

$$\underline{\mathbf{X}} = \underline{\mathbf{G}} \times_1 \mathbf{A}_1 \times_2 \mathbf{A}_2 \times_3 \mathbf{A}_3 + \underline{\mathbf{R}}, \tag{6}$$

where $\times_n$ is the mode-$n$ tensor-by-matrix product. $\underline{\mathbf{G}} \in \mathbb{R}^{R_1 \times R_2 \times R_3}$ is the core tensor and $\mathbf{A}_n \in \mathbb{R}^{I_n \times R_n}$ are factor matrices (Figure 1e). As a particular case, when core tensor is diagonal with $R = R_1 = R_2 = R_3$ this model is reduced to the CANDECOMP/PARAFAC or CP($R$) decomposition model (see Figure 1f), which has demonstrated to be very useful in a wide range of applications [47,48]. It is interesting to note that Equation (6) can be also written as a sum of rank-1 tensors as shown in Figure 1h which, if vectorized, it is then reduced to our general decomposition model of Equation (2) as follows:

$$\mathbf{x} = \sum_{i_1, i_2, i_3} g_{i_1 i_2 i_3} \boldsymbol{\phi}_{i_1 i_2 i_3} + \mathbf{r}, \tag{7}$$

where $\mathbf{x} = vec(\underline{\mathbf{X}})$, $\mathbf{r} = vec(\underline{\mathbf{R}})$, $g_{i_1 i_2 i_3}$ are coefficients and $\boldsymbol{\phi}_{i_1 i_2 i_3} = \mathbf{a}_3^{i_3} \otimes \mathbf{a}_2^{i_2} \otimes \mathbf{a}_3^{i_3}$ with $\mathbf{a}_n^i$ denoting the $i-$column of matrix $\mathbf{A}_n$. Tucker decomposition can provide data compression under two very different model assumptions leading to the following cases:

**Low-rank Tucker decomposition:** when the core tensor is much smaller than the original, i.e., $R_n \ll I_n$ [47,48] (see Figure 1e)

**Sparse Tucker decomposition:** when core tensor is of the same size or larger than tensor $\underline{\mathbf{X}}$ but it is sparse as illustrated in Figure 1g. In this case, by looking at Equation (7), we conclude that the Sparse Tucker model corresponds to the classical Sparse Coding model of (4) with a dictionary that is obtained as the Kronecker product of three mode dictionaries, i.e., $\boldsymbol{\Phi} = \mathbf{A}_3 \otimes \mathbf{A}_2 \otimes \mathbf{A}_1$ [49,50]. Mode dictionaries can be chosen from classical sparsifying transforms such as wavelets, cosine transform and others or, if enough data is available, they can be learned from a dataset, which usually provides higher levels of sparsity and compression. A Kronecker dictionary learning algorithm was introduce in [50] and later a variant with orthogonality constraints was proposed in [51].

### 2.6. Comparison of Methods for ML with Low-Quality Datasets

In Table 1, we summarize all the methods discussed in this paper and compare them in terms of their main characteristics, shortcomings, advantages and main applications. A detailed reference to the sections of this article where each of the methods is presented and discussed is given. Furthermore, main bibliographic references are indicated for each of the approaches.

**Table 1.** Comparison of methods for Machine Learning (ML) problems with low-quality datasets. Article sections in which these methods are discussed are noted in the first column and relevant references are included in the last column.

| Method | Characteristics | Shortcomings | Advantages | Application | References |
|---|---|---|---|---|---|
| **Class preserving transforms** (Section 1.1.1) | Ad-hoc; mostly images oriented but some extensions to other types of data were explored | Limited theory available; difficult to apply to arbitrary type datasets | Easy to use; widely available in deep learning platforms | Data augmentation | [7–17] |
| **Empirical Mode Decomposition (EMD) based data generation** (Sections 2.4, 3.1.2, 3.1.3 and 3.2) | Ad-hoc; based on the manipulation and recombination of Intrinsic Mode Functions (IMFs); | Lack of theoretical ground | Easy to use; capture dataset discriminative features; denoising power | Electroencephalography (EEG)/ invasive EEG (iEEG) data augmentation and denoising | [45,52–55] |
| **Transform domain based data generation** Section 3.1.3) | Ad-hoc; based on the manipulation and recombination of spectrum domain components obtained by Discrete Cosine Transform (DCT), Wavelets, etc. | Lack of theoretical ground | Easy to use; capture dataset discriminative features | iEEG data augmentation | [45,52,53,56] |
| **Statistical imputation** (Section 1.1.2) | Preprocessing step in ML; exploit statistical properties of datasamples; wide variety of methods, from simple ones (mean) to more sophisticated (regression, k-Nearest Neighbor (kNN), Self Organization Map (SOM), etc.) | Does not use the class label information of data samples | Computationally efficient | ML with incomplete or corrupted data | [18–23] |
| **Probabilistic modelling** (Section 1.1.2) | Gaussian Mixture Model (GMM) as data model; Bayesian classification; fitting model and classifiers in an Expectation-Maximization (EM) fashion; can be adapted to deep neural networks | Computationally expensive | Incorporates class label information of data samples; elegant theoretical approach | ML with incomplete or corrupted data | [19,24,28] |
| **Low-rank matrix completion** (Section 1.1.2) | Based on Singular Value Decomposition (SVD) | Computationally very expensive; not suitable for complex boundary functions | Incorporates class label information of data samples | ML with incomplete or corrupted data | [25–27] |
| **Tensor decomposition (TD) based imputation** (Sections 2.5, 3.1.1 and 3.3) | Preprocessing step in ML; based on low-rank TDs (e.g., Tucker, CANDECOMP/PARAFAC (CP), etc.) or sparse TDs | Does not use the label information of data samples | Exploits intricate relationship among modes in multidimensional data | ML with incomplete or corrupted data | [50,57–62] |

## 3. Results

This section illustrates the application of the methods discussed above and presents the results obtained in the following selected case studies: brain signal classification with missing, corrupted or small datasets (Section 3.1); classification of noisy faces (Section 3.2) and analysis of water network data (Section 3.3).

### 3.1. Brain Signal Classification

Brain signal activity can be recorded non-invasively by using, for example, electroencephalograpy (EEG) or invasively through electrodes located in the brain (iEEG). Decoding brain activity have found many applications in medicine and has great potential for the next generation of human-machine communication technologies. In a Brain Computer Interface (BCI) application, a user generates specific brain activity patterns that can be decoded by a machine learning algorithm. One popular paradigm in BCI is Motor Imagery (MI), which states that the brain activity generated by a subject imagining movement of one of their limbs for a few seconds activate areas in the motor cortex, which are similar to those that are activated with real movements, thus this particular neural activity can be detected by a machine learning algorithm (classifier).

#### 3.1.1. BCI with Missing/Corrupted Measurements

In BCI applications, noisy or missing data can arise. This can occur due to the lack of connection between wireless EEG headset and the computer, or because artifacts appear due to muscle movements, eye movements or electromagnetic interference, among others. Since EEG measurements can be organized as multidimensional datasets, in [57] a tensor completion approach was proposed, which consists in fitting a tensor decomposition model to the available clean measurements and then infer the noisy or missing parts based on those models (see Figure 2a). The advantage of using tensor methods, compared to classical interpolation algorithms, lies in the ability of these models to handle multidimensional information, in other words, they can capture the intricate relationship among entries in a multidimensional signal. For example, to infer a missing entry in an EEG data tensor, these methods can efficiently exploit the available information in other channels, time samples and trials.

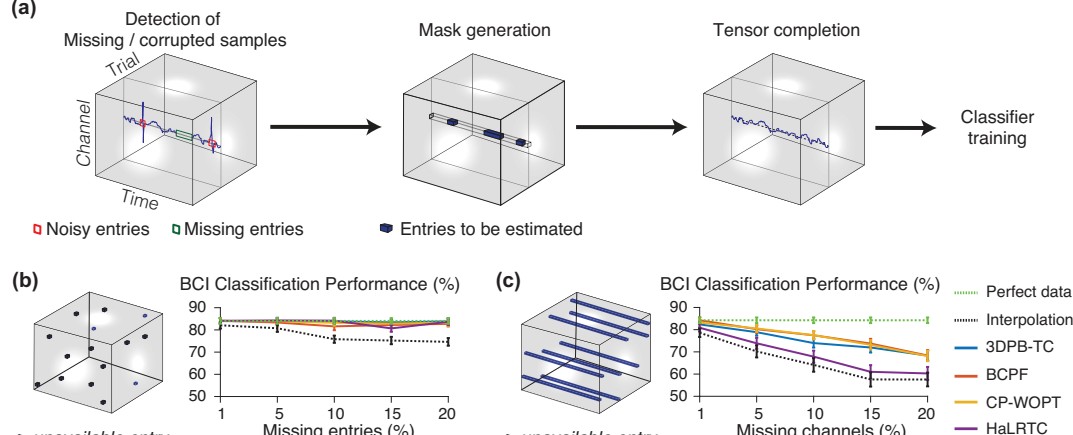

**Figure 2.** Training a BCI classifier (LDA) with noisy/missing EEG measurements. (**a**) Preprocessing steps: first, the positions in which the data is missed/corrupted are identified; then, a mask is created to ignore values in those positions; and finally, the tensor model reconstructs the missing data. (**b**) Results with randomly missing entries. (**c**) Results with random missing channels. (Figure adapted from [57]).

Several tensor decomposition models and tensor completion algorithms were compared on a freely available dataset (http://mon.uvic.cat/data-signal-processing/software/) in [57], which are

based on the CP model of the whole tensor, such as the CP Weighted Optimization (CP-WOPT) [58], the High accuracy Low Rank Tensor Completion (HaLRTC) [59] and the Bayesian CP factorization (BCPF) for tensor completion [60]; and one method that uses the Sparse Tucker decomposition of every $6 \times 6 \times 6$ tensor patch (subtensors), the 3D Patch-based Tensor Completion (3DPB-TC) [50]. In the latter case, a Kronecker dictionary was first learned from a clean EEG training dataset. For the experiments with incomplete measurements, two different patterns of missing data were considered: random missing entries, and random missing channels, as shown in Figure 2b,c, respectively. The latter case, represent a realistic situation in which some electrodes are simultaneously disconnected during a complete trial.

The performance of the tensor completion algorithms was compared, together with a simple interpolation strategy as shown in Figure 2b,c. The classification accuracy of imagined movement in a BCI experiment, using a Linear Discriminant Analysis (LDA) classifier, was evaluated in the perfect case (no missing data) and with each one of the recovered missing data through the tensor completion algorithms. Experimental results demonstrated that all tensor completion algorithms were able to recover missing samples increasing the classification performance compared to a simple interpolation approach, because tensor methods are able to exploit the multidimensional correlation of data. As expected, the random missing samples was easier to reconstruct using the neighbour points, while the random missing channels was more difficult because the amount information was missed in the same neighbourhood (the same channel, in that case). Therefore, as experimentally demonstrated, tensor completion algorithms could be used in real BCI data to avoid discarding noisy frames or frames with missing data, and instead recover the missing data using that technique.

### 3.1.2. Efficient Data Augmentation for BCI

Small datasets are common in many EEG applications. This is especially the case when developing systems for automatically detect some brain disease or brain injuries. Basically, because sometimes it is not easy or just impossible to have enough patients from which to record EEG or iEEG signals. MI-BCI systems for neurorehabilitation require a calibration step before being used. This is due to the fact that the system needs a classifier which is particular for each subject and session because, for example, the location of the electrodes in each session will never be exactly the same. The Common Spatial Pattern (CSP) algorithm [63] is habitually used to extract features. This calibration step implies recording several MI frames which will be used to extract the CSP filters and train the classifier. Each frame is composed by the EEG recordings of a particular trial.

Since the quality of the classifier can be greatly improved by using a large number of frames from each type of MI [63], it is habitual to record 100 or more frames, in total. Taking into account that the MI paradigm can last about 10 s per frame, approximately a minimum of 16 min will be employed in the recording session. Over that, the time needed to set up the EEG montage has to be added.

A way to shorten the calibration time is by reducing the number of registered frames but generating artificial ones to keep high the total number of frames. While available data augmentation techniques are proved to be efficient to boost the training of neural networks and support vector machines, they where developed mainly for image datasets, where natural transformations are croping and rotations, for example. These type of transforms have no sense for EEG data and new methods for data augmentation need to be developed. Methods based on signal decomposition and recombination of its main components are a natural way to solve that issue. In [52], the authors developed, for the first time, a method to generate artificial frames based on an EMD decomposition/combination strategy. Starting from a real frame collection, a new artificial frame of a specific class is built as described in Figure 3, comprising the following steps:

1.　Randomly select $N$ frames from the set of frames belonging to the selected class.
2.　Decompose, using EMD, each one of the $N$ frames, generating a set of IMFs per channel and frame.

3. Then, select the first IMF from the first selected frame (one per channel and keeping the same position for each channel), the second IMF from the second selected frame, and successively until the $N$th frame, which contributes with its $N$th IMF.

4. Add up all the IMFs corresponding to the same channel to build each new EEG channel of the new artificial frame.

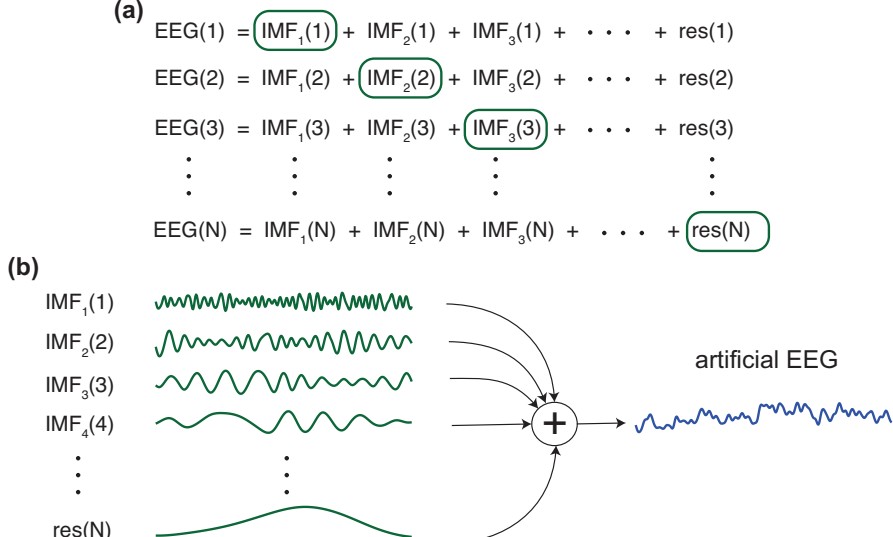

**Figure 3.** EEG data augmentation: (**a**) For each new EEG signal to be generated, $N$ available EEG signals are randomly selected and their EMDs are computed. (**b**) To generate an artificial EEG signal, IMFs from different signals are combined.

Using this method, authors in [52] were able to diminish the amount of acquired data for the calibration step in a BCI scenario while maintaining the performance. Specifically, they replaced original frames with artificial frames and tested the behaviour of the classifier derived from the data. Depending on the percentage of artificial frames in the data, they concluded that up to 50% of the original frames could be replaced without affecting the classifier's performance as it is presented in Table 2. The performance of each classifier was validated trough the median absolute deviation (MAD) method to detect outliers [64], and the dispersion ratio $R$ was calculated as

$$R = |(x - \tilde{x})/MAD|, \tag{8}$$

where, for a set of measures, $\tilde{x}$ is the median value and $x$ is the measure to be tested. In the experiments, the error rate of the classifier was tested. Usually, if $R < 3$ the measure $x$ is not considered to be an outlier, i.e., this classifier had a similar behaviour even if a specific percentage of real frames was replaced by artificial frames. We can see in Table 2 that four subjects (S01, S04, S05 and S07) have a value of $R < 2.0$ for both left and right sides (motor imagery of left and right arm movement, respectively), and only one subject, S02, had a value of $R > 3.0$ for the right side movement imagination.

**Table 2.** Dispersion ratio *R* computed in seven subjects (S01-S07) with Equation (8) for right (R) and left (L) classes at different levels of used artificial frames (AF). Results with $R > 3$ are highlighted in red and with $2 < R < 3$ in orange.

| | S01 | | S02 | | S03 | | S04 | | S05 | | S06 | | S07 | |
|---|---|---|---|---|---|---|---|---|---|---|---|---|---|---|
| AF(%) | R | L | R | L | R | L | R | L | R | L | R | L | R | L |
| 2.5 | 0.12 | 0.67 | 0.22 | 0.64 | 0.58 | 1.27 | 0.32 | 0.31 | 0.32 | 0.27 | 0.33 | 0.64 | 0.34 | 0.69 |
| 5.0 | 0.05 | 1.03 | 0.82 | 0.56 | 1.11 | 1.02 | 0.46 | 0.45 | 0.18 | 0.35 | 0.47 | 0.83 | 0.01 | 0.63 |
| 7.5 | 0.29 | 0.88 | 1.03 | 0.07 | 1.06 | 1.51 | 0.51 | 0.51 | 0.00 | 0.02 | 1.17 | 1.49 | 0.46 | 0.62 |
| 10.0 | 0.37 | 1.13 | 0.99 | 0.11 | 1.19 | 1.75 | 0.80 | 0.46 | 0.38 | 0.08 | 1.04 | 1.66 | 0.49 | 0.84 |
| 12.5 | 0.24 | 0.94 | 1.42 | 0.04 | 1.89 | 1.86 | 1.00 | 0.44 | 0.46 | 0.27 | 0.87 | 1.52 | 0.40 | 0.85 |
| 25.0 | 0.09 | 1.44 | 2.79 | 0.44 | 2.13 | 1.94 | 1.28 | 0.61 | 0.96 | 0.78 | 0.71 | 2.09 | 0.51 | 1.28 |
| 37.5 | 0.11 | 1.55 | 3.12 | 0.41 | 1.97 | 2.01 | 1.20 | 0.69 | 1.07 | 1.18 | 0.57 | 2.66 | 0.73 | 1.92 |
| 50.0 | 0.15 | 1.45 | 2.86 | 1.00 | 2.18 | 2.68 | 1.27 | 1.06 | 1.42 | 1.23 | 0.62 | 2.76 | 0.73 | 1.86 |

Another application based on the same idea, but for deep neural network classifiers purposes, was proposed in [53]. Here, the authors also used a BCI scenario to exemplify how the EMD decomposition/recombination technique could be useful to create enough artificial data to train a deep learning structure while avoiding overfitting. This is a very important application when dealing with small datasets. If the classifier has many parameters, which is what happens in a deep learning structure, the system can easily suffers from overfitting. The only way to reduce it is by simplifying the classifier, hence, for example changing the deep learning structure for a more traditional machine learning structure with few parameters, or increasing the amount of data. When data is difficult to acquire or impossible for any reason (economical, practical, availability, etc.), the EMD decomposition/recombination technique can be used to generate artificial data. This is what was explored in [53]. In this work, a convolutional neural network and a wavelet neural network were proposed to classify BCI data. To be able to train deep learning structures with few data, the method described above was used as a data augmentation strategy. The authors showed experimentally that the artificial EEG frames were useful to improve the training of neural networks.

### 3.1.3. Epileptic Focal Detection with Limited Data

Epilepsy is a general term for a condition that causes repetitive seizures caused by excessive activity of neurons in the cerebrum, and such excessive activity is detected on EEG tests. Focal epilepsy, in which a part of the brain becomes abnormally excited and causes seizures, may be reduced or cured by removing the abnormally excited part of the brain (epileptogenic zone). Identification of the epileptogenic area requires intracranial electroencephalography (iEEG), in which electrodes are placed intracranially and measured, as well as brain imaging evaluation such as MRI.

The identification of the epileptogenic zone requires a long-term iEEG recording. Besides, the number of clinical experts (epileptologist) is limited. Therefore, the automated identification of the brain area of seizure onset of focal epilepsy (a.k.a automated focal identification) using interictal (non-seizure) iEEG signals is in strong demand. However, clinical iEEG data usually stay in each medical facility and cannot be in public, so that the amount of data available for training a machine learning model is also limited. This situation brings the necessity of appropriate processing for a small dataset.

A possible and straightforward way to cope with this problem is data augmentation. Two types of approaches to the iEEG data augmentation have been proposed recently. The first method is to augment data in the signal domain [54]. The second method is the data augmentation in the feature domain [56]. Both methods work efficiently in identifying focal locations from interictal iEEG.

The data augmentation in the signal domain applies orthogonal transforms such as the discrete Fourier transform, the discrete cosine transform (DCT), and the discrete wavelet transform (DFT) to the raw iEEG signal, and then the transform coefficients are shuffled across multiple epochs (or segments). Zhao et al. constructed an efficient and sophisticated method for the data augmentation based on the DCT [54] and applied to the Bern-Barcelona dataset [65], which is a well-known dataset

consisting of epileptic iEEG signal at focal and non-focal locations. The steps to increase the number of samples are summarized as follows:

1.  Randomly choose seven iEEG signals from the dataset and apply the DCT to obtain the spectrum.
2.  Segment the spectrum into the seven physiological frequency bands (Delta: 0–4 Hz, Theta: 4–8 Hz, Alpha: 8–13 Hz, Beta: 13–30 Hz, Gamma: 30–80 Hz, Ripple: 80–150 Hz, and Fast Ripple: 150–Nyquist Hz), extract one frequency band of each of the decompositions, from lowest to highest frequencies, and merge the seven extracted components (frequency bands) to create a new artificial spectrum. For example, we can extract the delta, the theta, the alpha, the gamma, the ripple, and the fast ripple from the first, the second, the third, the fourth, the fifth, the sixth, and the seventh signal, respectively.
3.  Apply the inverse DCT to the artificial spectrum in the frequency domain to obtain an artificial signal in the time-domain.

This is illustrated in Figure 4. The above procedure should be applied to focal and non-focal signals separately. Since this approach to the data augmentation increases the number of samples in the signal-domain, it may be suitable for deep learning-based techniques. In the work [54], the authors successfully applied this data augmentation for a convolutional neural network to identify the focal signals. The data augmentation strategy demonstrated to be useful: using 20% of available real data plus artificial data, the classifier was able to improve its accuracy results to 83.91%, compared to 81.52% obtained using only real data.

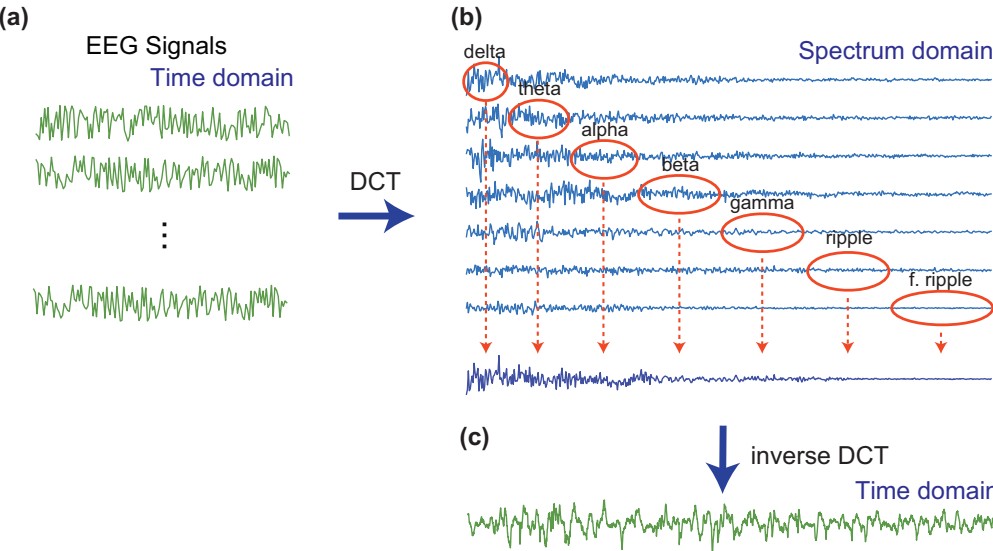

**Figure 4.** An artificial signal generation with the DCT. (**a**) Seven intracranial iEEG signals at either focal or non-focal area. (**b**) DCT coefficients in the spectrum-domain. The spectra are segmented into seven physiological sub-bands, and the sub-band components extracted from each spectrum are merged to create an artificial spectrum. (**c**) The inverse DCT leads to the resulting artificial signal.

The other approach to data augmentation is a method in the feature-domain. Most of the classification methods for epileptic EEG signals typically extract features from the raw signals. In general, a conventional yet effective classifier, such as an SVM, requires fewer samples than a deep neural network. Thus, increasing the number of samples in the feature-domain is a strategy to balance positive and negative samples. In the case of epileptic iEEG data, positive (focal) samples are much fewer than negative samples, and this imbalance data can deteriorate the performance of classification accuracy. Akter et al. [56] successfully applied data augmentation based on an adaptive synthetic oversampling approach (ADASYN) [66] to balance an in-hospital dataset of epileptic interictal iEEG signals to identify the seizure onset zones automatically.

### 3.2. Classification of Noisy Faces

Working with noisy data is a challenge and generates many problems when developing classification systems. Denoising algorithms have to be applied before deriving the classifier and /or when using it. This can be the case in image processing, in which noise can corrupt the image. In an image recognition or verification system, noise could make the task difficult inducing more errors. This is why robust to noise systems are needed. To this end, several strategies can be implemented, for example using classical filters such as a Gaussian [67], bilateral [68], arithmetic [69], median [70] or Wiener filters [71]. However, all of these filters are good in some situations or type of noise, but not good enough in others.

To overcome that issue, a denoising technique based on an empirical mode decomposition with Green's functions was proposed in [55]. The system uses the capability of the bi-dimensional EMD decomposition to capture (almost all of) the noise in the first IMFs. Therefore, the noisy images are decomposed using a bi-dimensional EMD algorithm, then the first modes are eliminated and the rest of the modes are summed up together to recover the remaining image, almost without noise. In this specific work, the bi-dimensional EMD algorithm was the Green's function in tension BEMD (GiT-BEMD) which uses Green's functions to interpolate the surface of the images [72].

The method used in [55] is depicted in Figure 5, in which we can see that the image is decomposed by means of the GiT-BEMD algorithm, the first IMF is discarded, the image is later reconstructed without the contribution of the noise and used to feed a classifier.

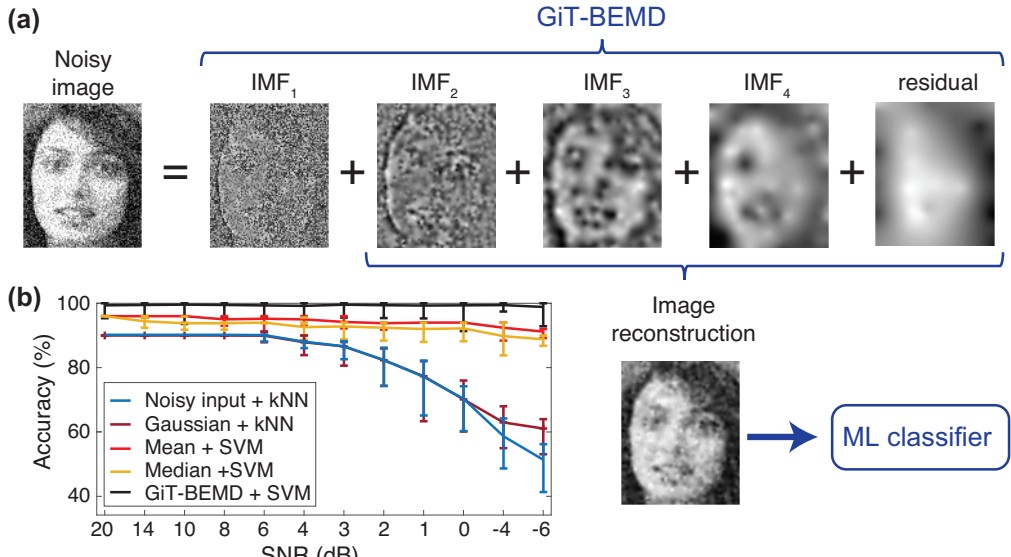

**Figure 5.** (**a**) The new proposed approach to eliminate the noise and improve the classification accuracy is based on the GiT-BEMD decomposition. The high frequency IMFs are discarded and the (noiseless) image is reconstructed by summing up the rest of the modes. This is the image that will feed the classifier. (**b**) Comparison of classification results using a Support Vector Machine (SVM) and K-Nearest Neighbor (kNN) classifiers applied to noisy, filtered faces (Gaussian, Mean, Median) and GiT-BEMD processed faces.

Experiments were carried out using several type of noise (Gaussian, Uniform, Laplacian and Speckle) and at several levels (SNR from $20_{dB}$ to $-6_{dB}$). The classification accuracy was compared to the ones obtained with classical filters (the ones named before). Classical filters were able to keep good performance when $SNR > 3_{dB}$, but they started to fail in much noisy scenarios, corresponding to $3_{dB} > SNR > -6_{dB}$. The proposed method was able to maintain the same level of performance in all the ranges, from almost noise-free images ($SNR = 20_{dB}$) to highly noisy images ($SNR = -6_{dB}$). Similar results occurred in the verification case, in which the Equal Error Rate (EER) were reported to

evaluate the results. Independently of the type of classifier used (SVM or k-NN), GiT-BEMD method obtained the lowest EER.

Finally, it is interesting to note that the method is efficient for any type of noise and under high levels of it, and is transparent to the user, hence simple to apply. There are no parameters to tune because the GiT-BEMD algorithm is data-driven, and the IMFs of the images are obtained automatically. Bidimensional EMD algorithms, and specially GiT-BEMD algorithm, are able to decompose images in several IMFs, capturing the noise in the first IMF, which allows for training a classification system that is robust to noisy scenarios.

### 3.3. Scada Data Completion in Water Networks

In industrial applications, usually having a Supervisory Control And Data Acquisition (SCADA) system collecting data, is habitual to have missing values due to several problems (sensor failures, communication loss, etc.). In this scenario, tensor completion algorithms can be used to reconstruct missing data providing a better alternative compared to classical interpolation methods. Because of the cyclic behaviour of data consumption, a tensor structure can be defined considering time scales (days and weeks). The redundancy present into the data is exploited by the tensor completion algorithms, allowing to recover missing data with greater accuracy outperforming classical interpolation or filtering methods.

Drinking water network distribution enterprises usually have a SCADA system which manages the information collected by flow-meters, manometers, level sensors, valves, pumps, etc. By accessing to this centralized dataset, they are able to manage and optimize the operation of the water network.

One of the major problems involved in managing long term SCADA data is to deal with the loss of bursts of data due to sensor failures, sensor re-calibrations, or communication failures that take time to be repaired and therefore cause the loss of entire bursts of data. Completing the data lost in bursts remains a difficult task, and most data completion methods that work fairly well when data are lost more or less evenly distributed over time, collapse in that situation.

Water network data completion was investigated in [61,62]. For this application, authors used data coming from the drinking water network distribution enterprise named Aigües de Vic S.A. (AVSA) which is the responsible for the water supply of the city of Vic, in Catalonia, where the heterogeneous data from the SCADA system are stored every 5 min in Structured Query Language (SQL) databases. To explore how tensor completion methods would behave in this scenario, a block of 77 consecutive weeks of data was selected. Then, some parts of the data were deleted to simulate missing burst. The Mean Square Error (MSE) per sample was used as a measure to check the accuracy of the reconstructed data.

In [61], besides of taking advantage of the classical methods, an additional improvement was achieved by performing a tensorization of the data and applying tensor decomposition to recover missing data. This tensorization operation is illustrated in Figure 6.

In [62], a new approach was developed that improved the performance of the previous method by performing two concatenated tensor decompositions. The new approach has several steps. The first step, consisted of applying a smoothing process to the signal to avoid oscillations between adjacent discrete values eventually produced around the point of quantification. The second step consisted of using a very rough imputation method (linear interpolation). After this operation, no empty values were present. Then tensorization that places the original burst positions precisely in the central positions, the burst centered tensorization, was performed (Figure 6a). The tensor obtained was used to make a low-rank tensor model to capture the background trend of the missing burst. The data of the low-rank model in the burst positions obtained was offset corrected and burst-centred again in a new tensor which was used to find a most refined model by employing more modes in a second tensor approximation. The samples $\hat{x}_i$ occupying the positions of the burst were retrieved, offset corrected and became the output of the algorithm. The concatenated decompositions that optimize the results independently of the length of the burst to recover were the Tuker(4,6,1) followed by the

Tuker(4,7,7) when using the Tucker model, and the CP(1) followed by the CP(15) when using the CANDECOMP/PARAFAC model. Table 3 shows a comparison between the best classical algorithm tested, the one based on the combination of forward and backward predictors, the singleDecomp [61] and the doubleDecomp [62] algorithms. TK (Tucker) and CP (CANDECOMP/PARAFAC) indicate the decomposition model used by the algorithms. The comparison is carried out considering two tensor dimensions and two lengths of lost bursts.

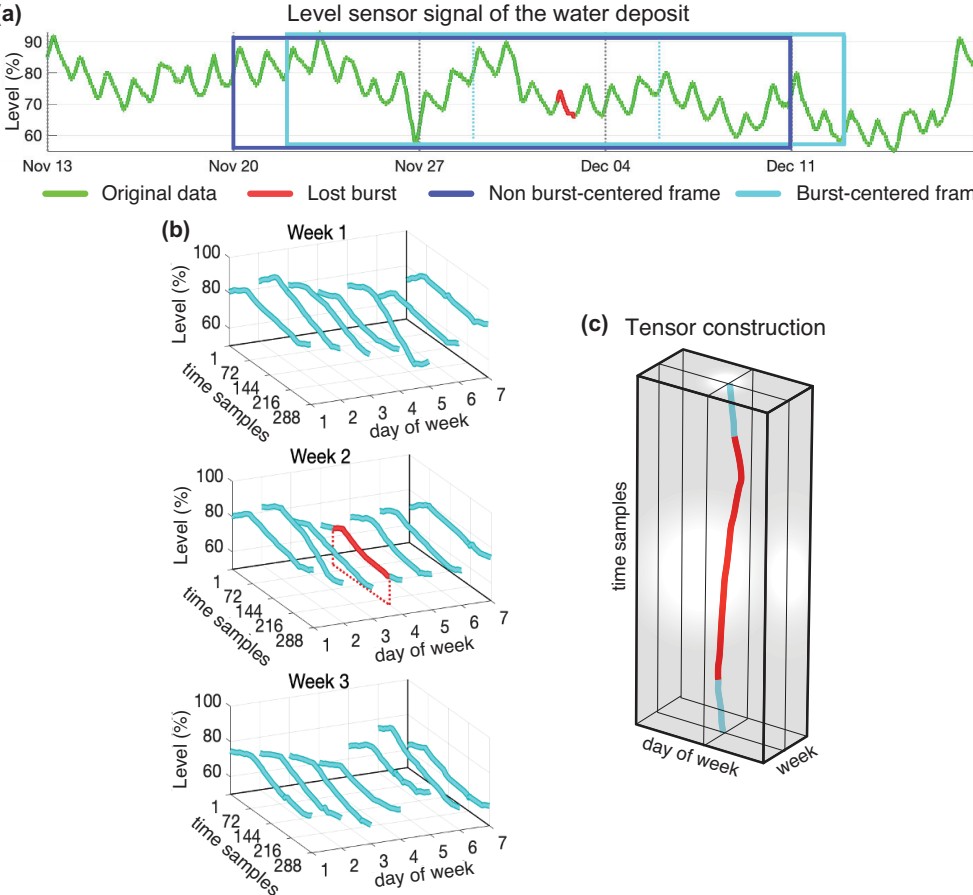

**Figure 6.** Data tensorization of a 3 week tensor with 200 samples of lost data bursts. In (**a**) the green line shows the original data, and the red line shows the lost burst. The soft blue window shows the data introduced in burst-centered tensor, which forces the burst to be in the center of the window. Panels (**b**) shows how the continuous flow of data in the soft blue window is fragmented to be allocated in the tensor as shown in panel (**c**).

**Table 3.** Algorithms' performance in terms of the MSE per sample.

| Method | Weeks | MSE/Sample | |
|---|---|---|---|
| | | Burst Length = 100 | Burst Length = 200 |
| Forward & Backward Predictors | - | 1.11 | 2.23 |
| SingleDecomp—CP | 3 | 0.87 | 1.78 |
| SingleDecomp—CP | 7 | 0.80 | 1.58 |
| SingleDecomp—TK | 3 | 0.80 | 1.43 |
| SingleDecomp—TK | 7 | 0.71 | 1.28 |
| DoubleDecomp—CP | 3 | 0.55 | 1.05 |
| DoubleDecomp—CP | 7 | 0.52 | 1.02 |
| DoubleDecomp—TK | 3 | 0.55 | 1.04 |
| DoubleDecomp—TK | 7 | **0.50** | **0.97** |

Best results are indicated in bold text.

## 4. Conclusions and Discussion

Machine learning methods were typically designed by assuming that training data is perfect and of infinite size. However, in real life, machine learning practitioners need to deal with imperfect training data or datasets of limited size. To alleviate the problem of incomplete/corrupted datasets or to increase the size of a training dataset, it is necessary to use powerful data models that can capture the essential features of the dataset. In this article, a unifying introduction to signal decomposition methods is presented, which are available in different flavors but sharing a common property: every signal in a dataset (sample) can be written as a linear combination of elementary, simpler components.

The main idea behind the reviewed decomposition models is that they are able to be learned from a limited or low-quality dataset. Once the right model for the dataset of interest is learned from the available samples, one can use it for different tasks: (1) to complete missing entries in data samples, (2) to compensate distortions or eliminate noise in data samples, and (3) to artificially create class-preserving new data samples.

We have demonstrated that low-rank, sparse coding and EMD decomposition methods are excellent candidates for models that can capture essential information of a dataset. All these methods can be applied to vector as well as to tensor datasets. However, some issues remain and there is space for improvement in future research. For example:

- The decomposition methods reviewed in this work for imputation of missing/corrupted values do not exploit the class label information in a supervised learning scenario. A possible further improvement of current methods is to incorporate label information into the decomposition models. We believe that missing data values could be better recovered if the class label of the corresponding data sample is known.
- EMD based data augmentation was developed in an ad-hoc fashion. We believe that more theoretical insights could be explored allowing future improvements, for example, by re-designing the way that IMFs are calculated in order to produce class-preserving artificial samples.

This review article illustrates the application of a variety decomposition methods to vector and tensor datasets in a wide range of technological areas including: classification of brain signals (EEG and iEEG), identification of face images and analysis of water network data. While this represents a non-exhaustive review of existing methods and applications of machine learning with low-quality datasets, we believe that it can be a useful reference for machine learning practitioners who are normally faced with incomplete, noisy or small datasets, and can inspire new methods to address these fundamental problems.

**Author Contributions:** Conceptualization, C.F.C., J.S.-C. and S.Z.; investigation C.F.C., J.S.-C., P.M.-P, S.Z. and T.T.; writing—Original draft preparation C.F.C., J.S.-C., P.M.-P. and T.T. All authors have read and agreed to the published version of the manuscript.

**Funding:** This research was funded in part by JST CREST Grant Number JPMJCR1784 and by the University of Vic—Central University of Catalonia (ref. R0947). J.S.-C. work is also based upon work from COST Action CA18106, supported by COST (European Cooperation in Science and Technology).

**Acknowledgments:** We are grateful to the anonymous reviewers for their valuable comments, which helped us to improve the first version of this manuscript.

**Conflicts of Interest:** The authors declare no conflict of interest.

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
