# Peer review of "Decomposition Methods for Machine Learning with Small, Incomplete or Noisy Datasets"

_applsci, doi:10.3390/app10238481_

Round 1
Reviewer 1 Report
The authors investigate a very interesting topic related to Machine Learning based on deep neural networks. They focuses on the emerging problem typical of the ML about the assumption that the datasets are “infinite”. The authors show a signal decomposition approach to improve the ML performances when the datasets are small or incomplete. However, in my opinion the manuscriprt should be considered as a "Review" (as indicated by the authors themselves in the abstract – line 5) because it does not introduce significant novelty, but they provide recent advances in literature. If the authors agree with my observation, I suggest the following points:
1) In the Introduction section the authors should describe more extensively the state of the art and provide more information on recent progress in the literature. In particular:
a) Line 37: the authors should describe in details the references [4-7], highlighting the progress recently made.
b) Lines 38-44: the authors should provide more information related to the practical applications [8-10] and they should also discuss the main results.
2) The References section should be expanded with more recent work."
Reviewer 2 Report
Authors of this manuscript found interest and presented a review for decomposition methods in ML, for small, incomplete and noisy data, and explained with some application scenarios, particularly for signal handling.
In general, the manuscript is easy to read, and properly-referenced. Notation and figures are illustrative.
Yet,
(1) although this manuscript is focused on decomposition methods, it would be better in the introduction, to briefly present some insights on other methods as well; Or at least cite or list some alternative solutions;
(2) It would be better if more insights can be presented in section 2; List of the method and present some concepts are hardly making up to a review, the readers can just go for the references for them; In-depth explanation with your insights are the parts where your contributions lay on;
(3) Similar to point (2), we expect a higher-level summary of the results, rather than adapt figures and tables from other works; Some results may be adapted, while we should focus more on your findings by reading those works; What is the shortcomings, and what is the merits? How can we improve them in future works? What is the direction of possible future works?
(4) We do expect a comparison between those methods, not only in performances but also from your understanding.
Round 2
Reviewer 1 Report
The authors have responded to my comments, I have no additional ones.
Author Response
Thanks again for your valuable comments that helped us to improve the manuscript.
Reviewer 2 Report
Most of my previous concerns are resolved to some extent, and I accept the authors' argument on self-contain and presentation on different applications.
Maybe it's my 4th comment on comparison not clear enough... thus, for a high-quality in-depth review, still I recommend authors present some side-by-side comparisons in terms of different methods, specifically for section 2, and for each method, it would better to present their characteristics, the shortcomings, and advantages, in what scenario should it be applied, and so on... you may even list it on a table or so if preferred.
Author Response
We thank the reviewer for clarifying the 4th comment on comparison of methods. We agree with the proposal. In the revised version of the manuscript, we have included a new subsection and a table with a summary of all methods discussed in the paper and a comparison of their characteristics, shortcomings, advantages and applications (see section 2.6 and Table 1, pages 8-9)